# *Drosophila* sessile hemocyte clusters are true hematopoietic tissues that regulate larval blood cell differentiation

Alexandre B Leitão[1]*, Élio Sucena[1,2]*

[1]Instituto Gulbenkian de Ciência, Oeiras, Portugal; [2]Departamento de Biologia Animal, Universidade de Lisboa, Lisbon, Portugal

**Abstract** Virtually all species of coelomate animals contain blood cells that display a division of labor necessary for homeostasis. This functional partition depends upon the balance between proliferation and differentiation mostly accomplished in the hematopoietic organs. In *Drosophila melanogaster*, the lymph gland produces plasmatocytes and crystal cells that are not released until pupariation. Yet, throughout larval development, both hemocyte types increase in numbers. Mature plasmatocytes can proliferate but it is not known if crystal cell numbers increase by self-renewal or by de novo differentiation. We show that new crystal cells in third instar larvae originate through a Notch-dependent process of plasmatocyte transdifferentiation. This process occurs in the sessile clusters and is contingent upon the integrity of these structures. The existence of this hematopoietic tissue, relying on structure-dependent signaling events to promote blood homeostasis, creates a new paradigm for addressing outstanding questions in *Drosophila* hematopoiesis and establishing further parallels with vertebrate systems.

## Introduction

In insects, the functions of hemocytes (blood cells) are very diverse and include phagocytosis, extracellular matrix deposition, AMP production, encapsulation, and melanization. Similarly to what happens in vertebrates, the different functions performed by insect hemocytes are, to some degree, compartmentalized into different cell types (*Honti et al., 2014*). Some mature blood cells retain the ability to divide when in circulation, but the majority of blood cell proliferation and differentiation occurs in the hematopoietic organs (*Grigorian and Hartenstein, 2013*). These organs provide the correct cellular and molecular environment for the control of cell proliferation and differentiation, namely in the so-called stem cell niches (*Koch and Radtke, 2007*; *Martinez-Agosto et al., 2007*). Thus, the study of hematopoietic organs structure and function is essential to understand how different mature blood cells arise and how their absolute and relative numbers are controlled.

In *Drosophila melanogaster*, embryonic hematopoiesis produces two different types of mature hemocytes: plasmatocytes and crystal cells. Plasmatocytes are phagocytic cells often functionally compared to vertebrate macrophages (*Evans et al., 2003*). Crystal cells are non-phagocytic cells known to produce prophenoloxidase, an essential component of the melanization cascade (*Binggeli et al., 2014*). Both plasmatocytes and crystal cells generated during embryogenesis persist into larval stages. Hemocytes in the larva can be found in circulation but the majority of them are attached to the cuticular epidermis as sessile cells (*Lanot et al., 2001*; *Kurucz et al., 2007b*; *Makhijani et al., 2011*). Hemocytes attached to the epidermis are not randomly dispersed but form stereotypical clusters of cells in every segment of the larva (*Zettervall et al., 2004*; *Makhijani et al., 2011*) indicating that some signal must direct hemocytes to these locations. In fact, it has been recently shown that peripheral nervous system (PNS) neurons attract hemocytes and provide unknown trophic molecules

*For correspondence: aleitao@igc.gulbenkian.pt (ABL); esucena@igc.gulbenkian.pt (ÉS)

Competing interests: The authors declare that no competing interests exist.

**eLife digest** Insects have several different types of blood cell, many of which are unable to divide to form new cells once they have matured. Instead, fresh blood cells are normally generated in specialized 'hematopoietic' organs, such as the lymph gland in the *Drosophila* species of fruit fly. The structure of these organs plays an important role in controlling how new blood cells develop.

*Drosophila* embryos make two types of blood cell: plasmatocytes and crystal cells. Both defend against harmful microorganisms, but in different ways. Plasmatocytes engulf and destroy invaders, whereas crystal cells release chemicals that encase microbes in a hardened gel. The blood cells made in the *Drosophila* embryo are still present once the fly enters its larval stages. At this stage of development, most of the blood cells are found in clusters attached to the cuticle that covers the larva's surface, but a few circulate freely around the larva's body.

As a *Drosophila* larva develops, the number of blood cells in the larva increases. However, previous work has shown these additional blood cells are not normally released from the lymph gland of the larva. Furthermore, mature crystal cells do not appear to form new cells by dividing in two.

Leitão and Sucena now show that the stationary clusters of blood cells produce new crystal cells in *Drosophila* larvae. Within the clusters, plasmatocytes are made to turn into crystal cells via a signaling pathway controlled by a protein called Notch. This pathway was already known to be essential for forming crystal cells. Leitão and Sucena also show that the structure of the clusters influences whether crystal cells are made, which means that the clusters can be considered to be hematopoietic tissue.

It is now important to compare how the production of the same cell type is controlled in two distinct hematopoietic structures: the clusters and the lymph gland. From this comparison, general principles may be drawn and tested in other systems, including vertebrates.

for their survival (*Makhijani et al., 2011*). The larva also possesses a hematopoietic organ, the lymph gland, where plasmatocytes and crystal cells differentiate from prohemocytes (*Crozatier and Meister, 2007*). Prohemocytes residing in the medullary zone of the lymph gland are instructed by cells from the posterior signaling center (PSC) to maintain their quiescent state or to differentiate into mature plasmatocytes or crystal cells (*Crozatier et al., 2004*; *Mandal et al., 2007*). During the differentiation process, it has been suggested that cells migrate and occupy the most cortical zone of the lymph gland (*Jung et al., 2005*; *Krzemien et al., 2010b*). An essential aspect of the *Drosophila* larval hematopoiesis is that hemocytes produced in the lymph gland do not disperse from the organ until pupariation or upon injury such as parasitoid wasp egg infection (*Holz et al., 2003*; *Honti et al., 2010*). Hence, in homeostatic conditions, differentiated hemocytes in the lymph gland do not contribute to the circulating and sessile hemocyte population. Nonetheless, the hemocyte population found in circulation and in sessile patches expands throughout larval development. Plasmatocytes are mitotically active cells (*Rizki, 1957*; *Lanot et al., 2001*) expanding during larval development by self renewal (*Makhijani et al., 2011*). On the other hand, all reports thus far concur in that mature crystal cells do not divide during larval stages (*Krzemien et al., 2010b*; *Lanot et al., 2001*; *Rizki, 1957*), although they have been shown to proliferate during embryogenesis (*Lebestky et al., 2000*). Further characterization of a yet unknown source and undetermined mechanism of crystal cell differentiation is required to understand how its number increases during larval development.

Although little is known on how crystal cells are formed outside the lymph gland, it has been shown that Notch signaling is necessary to form these cells (*Duvic et al., 2002*; *Lebestky et al., 2003*). In the lymph gland, the role of Notch signaling in crystal cell formation is cell autonomous (*Mukherjee et al., 2011*). Notch activation is sufficient in hemocytes to induce the expression of *lozenge*, the first known transcription factor in crystal cell development (*Lebestky et al., 2000*). One particularity of Notch signaling is that it requires cell contact since the two known *Drosophila* Notch ligands, Serrate and Delta, are membrane bound proteins (*Fiúza and Arias, 2007*). In the lymph gland, Serrate-positive hemocytes induce neighboring cells to adopt crystal cell fates (*Lebestky et al., 2003*; *Mukherjee et al., 2011*; *Ferguson & Martinez-agosto 2014*). Outside the lymph gland, only in sessile clusters may we observe hemocytes establishing stable cell–cell contacts between them (*Lanot et al., 2001*). In fact, hemocytes in clusters are densely packed and linked through interdigitations (*Lanot et al., 2001*),

particularly in the last two abdominal larval segments, the putative posterior hematopoietic tissue (PHT) (Kurucz et al., 2007).

Indeed, in recent years, the idea that hematopoietic properties must reside outside of the lymph gland has been put forward explicitly by the Andó laboratory (*Márkus et al., 2009*). Firstly, in a descriptive endeavor by Kurucz et al. (2007) where an operational posterior hematopoietic tissue (PHT) consisting of the sessile hemocyte clusters in the last two abdominal segments is postulated; and later in a set of experiments showing that hemocytes taken from these clusters have the ability to differentiate into lamellocytes upon transfer to a different larva (*Márkus et al., 2009*; *Honti et al., 2010*). Importantly, sessile hemocytes in clusters constitute the biggest compartment of hemocytes in the larva (*Lanot et al., 2001*; *Makhijani et al., 2011*), contained within epidermal and muscle tissue in a structure that has been called hematopoietic pockets (*Makhijani et al., 2011*). Moreover, the sessile plasmatocytes in such clusters have a higher division rate than those in circulation (*Makhijani et al., 2011*). However, to consider the hemocyte clusters as a *bona fide* hematopoietic tissue, evidence is needed that their structure is necessary to control cell proliferation and/or cell fate decisions.

In this study, we directly test the hypothesis that the hemocyte clusters constitute a hematopoietic tissue by addressing systematically the following questions: (i) are crystal cells differentiating in these clusters? (ii) is the structure/architecture of these clusters necessary for this function? and (iii) what is the role of the Notch pathway in this hematopoietic role?

## Results

### Crystal cell numbers increase during larval development through de novo differentiation

In homeostatic conditions, the embryonic-derived hemocyte population consists of plasmatocytes and crystal cells. It is possible to distinguish these two cell types with several combinations of cell markers (*Lebestky et al., 2000*) such as the two live genetic drivers: HemolectinΔ-nuclearDsRed (*Clark et al., 2011*) and Lozenge-GAL4 in combination with UAS-EGFP/mCD8GFP (*Lebestky et al., 2000*). With this combination of markers we can distinguish across the larval cuticle, $Hml^+Lz^-$ from $Hml^+Lz^+$ sessile hemocytes (*Figure 1A,A′*). Lozenge is the first marker known in the genetic cascade that leads to crystal cell differentiation and its expression is maintained as the cell matures (*Lebestky et al., 2000*). Hemolectin promoter has been used in different *Drosophila* transgenic lines to mark the majority of hemocytes (*Sinenko and Mathey-Prevot, 2004*; *Clark et al., 2011*). Hence, $Hml^+Lz^+$ cells are fully mature crystal cells or differentiating crystal cells while $Hml^+Lz^-$ cells are plasmatocytes. During maturation, crystal cells loose HmlΔ-GAL4 expression (*Mukherjee et al., 2011*). The same is observed with HmlΔ-nuclearDsRed but only in rare cells. This difference may be explained by the different degradation times of nuclearDsRed and cytoplasmatic GFP. In fact, it is possible to detect a higher proportion of $Hml^-Lz^+$ cells with HmlΔ-cytoplasmaticDsRed (*Figure 1—figure supplement 1*). Another characteristic that distinguishes plasmatocytes from crystal cells is that the former tend to increase in size as they mature (*Terriente-Felix et al., 2013*). Measuring hemocytes cell areas in the three different populations of cells, we can observe a clear difference in size distributions between $Hml^+Lz^-$, $Hml^+Lz^+$, and $Hml^-Lz^+$ cells (*Figure 1—figure supplement 2*).

When quantifying total hemocyte counts throughout larva development, it is undisputed that both plasmatocytes and crystal cell numbers increase (*Rizki, 1957*; *Lanot et al., 2001*; *Makhijani et al., 2011*). Here, we focus on third instar larvae because this is the developmental window in which the majority of larval hemocytes are originated (*Lanot et al., 2001*; *Makhijani et al., 2011*). Moreover, at this stage we could develop reliable in vivo imaging procedures that render our analysis and interpretations more pertinent (see below). It has been suggested repeatedly that during larval stages mature crystal cells are post-mitotic (*Rizki, 1957*; *Lanot et al., 2001*) making it reasonable to assume that new crystal cells differentiate as development proceeds. This can be achieved by inducing new crystal cell precursors, proliferation of crystal cell precursors or simply by maturation of precursor cells already present in the larval body cavity. The earliest known marker predictive of crystal cell differentiation is Lozenge and it has been reported that during embryogenesis $Lz^+$ cells can proliferate (*Lebestky et al., 2000*). We checked whether $Lz^+$ cells increase in number throughout third instar larval development or if their number is fixed and crystal cells mature from these precursors. We counted the total number of sessile hemocytes in the HmlΔ-nuclearDsRed LzGAL4>GFP/mCD8GFP larvae (see 'Materials and methods'). We could confirm that $Hml^+$ cells increase as third instar larval

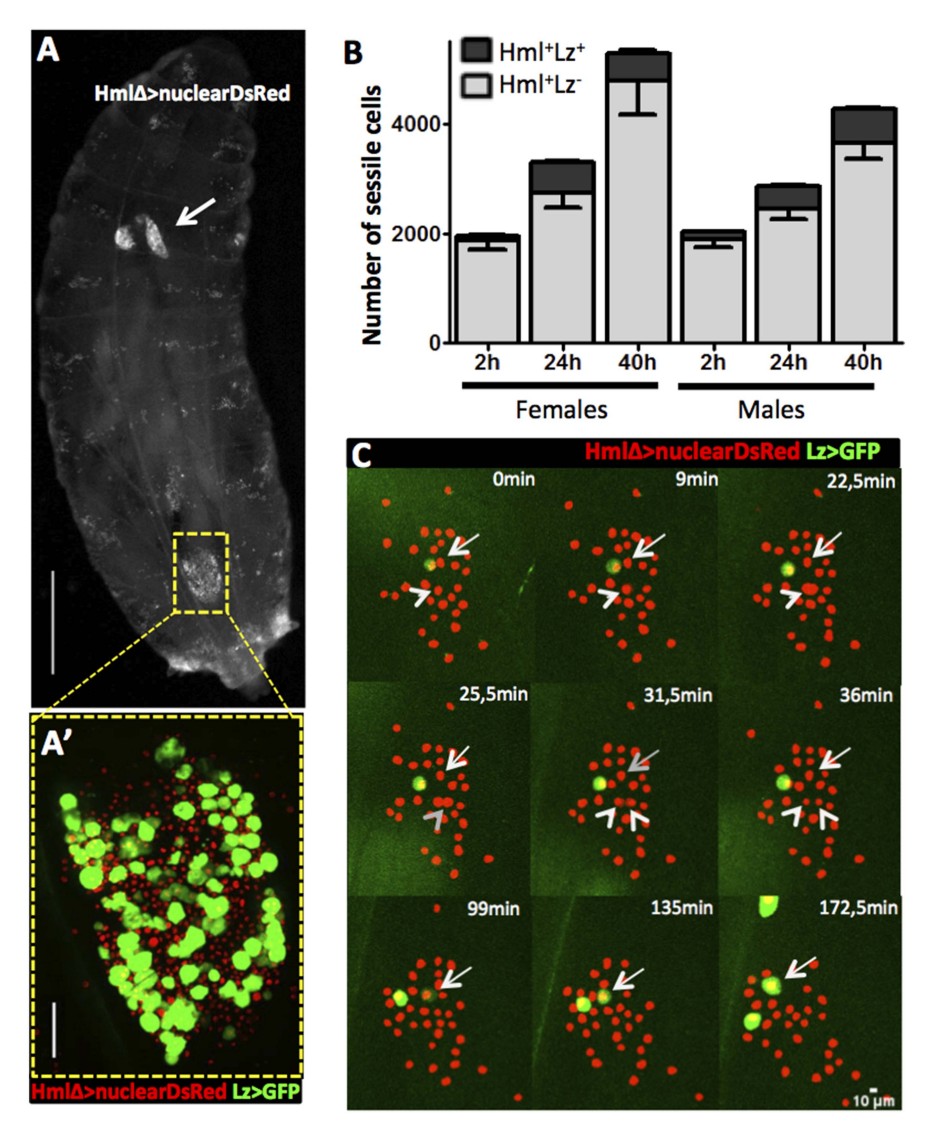

**Figure 1**. Hml⁺Lz⁺ cells increase during larval development by de novo differentiation. (**A**) Dorsal view of a third instar larva with hemocyte nuclei marked using HmlΔ-nuclearDsRed. The lymph gland (arrow) and sessile hemocytes along the body axis are visible, particularly in a big cluster on the A7 segment (square). Scale bar = 1 mm. (**A′**) Magnification of the A7 hemocyte cluster showing that it is constituted of Hml⁺Lz⁻ and Hml⁺Lz⁺ cells. Scale bar = 50 µm. (**B**) Throughout third instar larval development of both females and males, Hml⁺ sessile cells (grey bars) increase accompanied by an increment of sessile Lz⁺ cells (black bars), n = 10 per time point, error bars represent the SEM. (**C**) Still images of a 3-hr video showing hemocytes marked by HmlΔ-nuclearDsRed; Lz>EGFP/CD8GFP. It is possible to observe cell divisions in Hml⁺Lz⁻ (arrow heads) and GFP induction (arrows).

The following figure supplements are available for figure 1:

**Figure supplement 1**. Example of a sessile hemocyte cluster (abdominal segment A7) in a HmlΔ-DsRed; Lz>mCD8GFP larva. It is possible to observe small Hml^high Lz^low cells (arrows) and Hml⁻Lz⁺ cells (asterisks).

**Figure supplement 2**. Probability density plots for the different cell type sizes found in sessile clusters of HmlΔ-DsRed; Lz>mCD8GFP larvae (n = 8 samples).

**Figure supplement 3**. Throughout the 3-hr period covered in our videos, we can observe that GFP intensity in Lz⁺ cells increases, as measured by mean grey value of the cell at the beginning (0 min) and at the end (180 min) of the video.

*Figure 1. continued on next page*

*Figure 1. Continued*

**Figure supplement 4**. Quantification of GFP intensity and cell area of Lz$^+$ cells in hemolymph smears of HmlΔ-DsRed; Lz>mCD8GFP larvae, shows a strong positive correlation between cell size and GFP intensity.

development proceeds, both in males and females (*Figure 1B*). The number of Lz$^+$ cells also increases in the same time period (*Figure 1B*). In females there is no difference in the number of Lz$^+$ in the last 16 hr of development. With this, late third stage larva females have less crystal cells than males which is not common in the majority of fly stocks where females tend to have a higher number of crystal cells than males (see 'Results' below). Nevertheless, the results clearly show that committed crystal cells (Hml$^+$Lz$^+$) are increasing in number during third instar larval development, in parallel with an expansion of the plasmatocyte (Hml$^+$Lz$^-$) population.

Next, we wanted to distinguish if Lz$^+$ cells expand by cell proliferation or by de novo differentiation from Hml$^+$Lz$^-$ cells. A way to achieve this is to directly visualize hemocyte clusters with live time-lapse imaging and calculate the proliferation and differentiation rates for each cell type. To that purpose, we developed a method for imaging epidermal hemocyte clusters in live larvae for periods of 3 hr. HmlΔ-nuclearDsRed; Lz-GAL4>EGFP/mCD8GFP early L3 male larvae (<12 hr after L3 ecdysis) were selected and prepared for imaging (see 'Materials and methods'). As expected, it was possible to see the division of Hml$^+$Lz$^-$ cell nuclei (Arrows in *Figure 1C*). Throughout this 3-hr period, we estimate that ~7% of Hml$^+$Lz$^-$ cells divide (n = 13 videos with mean 50 Hml$^+$Lz$^-$ and 6 Hml$^+$Lz$^+$ cells per video). No case of Hml$^+$Lz$^+$ cell division was seen in any video. Since the number of Lz$^+$ cells per video is small, we analyzed an extra set of videos to check only for Lz$^+$ cell divisions (7 videos, 118 Lz$^+$ cells analyzed through a 3-hr period). Still, no Lz$^+$ cell division was observed. This suggests that Hml$^+$Lz$^+$ sessile cells in dorsal clusters divide at an extremely low frequency or are post mitotic cells. In contrast, we confirm that plasmatocytes (Hml$^+$Lz$^-$) proliferate within sessile clusters as previously reported (*Honti et al., 2010*; *Makhijani et al., 2011*). Notably, it was possible to see induction of GFP in GFP$^-$ cells, demonstrating that an Hml$^+$Lz$^-$ cell is turning into an Hml$^+$Lz$^+$ cell (Arrowhead in *Figure 1C*) (see *Video 1*).

During our analysis cells that start with low expression tend to increase it with time (*Figure 1—figure supplement 3*). When we bleed larvae and analyze Hml$^+$Lz$^+$ cells, we also see a tendency for larger cells to have more GFP, measured by mean grey value of the picture (*Figure 1—figure supplement 4*). If larger cells had the same GFP signal they would have a lower mean grey value because the signal would be diluted. In conjugation, these results indicate that *lozenge* expression starts in Hml$^+$ Lz$^-$ cells morphologically indistinct from plasmatocytes and increases together with cell size as crystal cell maturation progresses.

During the first 24 hr after third instar ecdysis Hml$^+$Lz$^-$ increase ~1.4× while Hml$^+$Lz$^+$ increase ~2.7× (see *Figure 1B*). Since we do not observe Lz$^+$ cell divisions, it is important to check if the Hml$^+$Lz$^-$ to Hml$^+$Lz$^+$ differentiation measured ratio is sufficient to explain this increase during development. The proportion of Hml$^+$Lz$^+$-induced cells in our videos is ~3.5% in 3 hr. Knowing the differentiation rate of Hml$^+$Lz$^-$ into Hml$^+$Lz$^+$, we can extrapolate the number of Hml$^+$Lz$^+$ cells differentiated from Hml$^+$Lz$^-$ cells at given time (see 'Materials and methods'). The differentiation rate calculated in our video analysis is

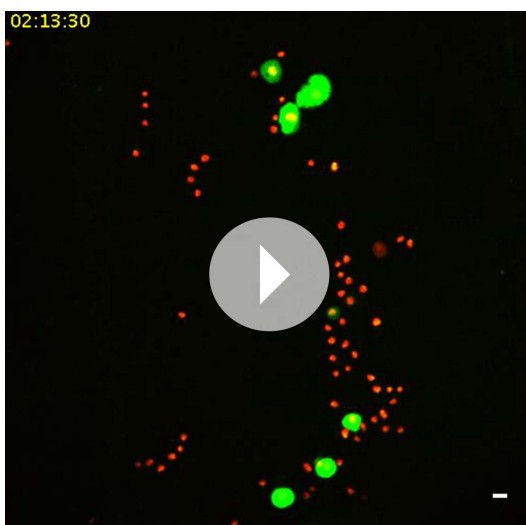

**Video 1.** Induction of lozenge expression in hemocyte clusters. HmlΔ-nuclearDsRed; Lz>EGFP hemocytes in a dorsal cluster. Examples of Hml$^+$Lz$^-$ hemocytes in division are highlighted with green circles and examples of Hml$^+$Lz$^-$ hemocytes differentiating into Hml$^+$Lz$^+$ hemocytes identified by a red circle. Scale bar = 10 μm.

sufficient to explain the increase in Hml$^+$Lz$^+$ cell number observed in the first 24 hr of third instar larval development. The mean increase of Lz$^+$ cells measured during development is 256.4 cells (*Figure 1B*) and the extrapolation gives an increase of 447 Lz$^+$ cells. On the whole, the analysis of these videos shows that Hml$^+$Lz$^+$ cells differentiate from Hml$^+$Lz$^-$ cells in hemocyte clusters at a rate that is sufficient to explain the increase in this cell type observed in the first 24 hr of L3 development.

## Plasmatocytes transdifferentiate into crystal cells

Only a small proportion of Hml$^+$Lz$^-$ cells become Hml$^+$Lz$^+$ in the course of third instar larval development. An important element to clarify is whether all Hml$^+$Lz$^-$ cells have the capacity to become Hml$^+$Lz$^+$ cells or whether this property is exclusive of a subpopulation of Hml$^+$Lz$^-$ cells. In the time window of interest, hemocytes can be divided into two sub-populations, NimrodC1$^+$ and NimrodC1$^-$ (Kurucz et al., 2007; *Csordás et al., 2014*). During lymph gland development, Nimrod is only detectable in mature plasmatocytes and it is not expressed in Lz$^+$ cells (*Terriente-Felix et al., 2013*; *Ferguson & Martinez-agosto 2014*). Thus, in the lymph gland Lz$^+$ cells differentiate from Hml$^+$ Nimrod$^-$ cells. To test if Nimrod$^-$ hemocytes in sessile clusters are the precursors of Lz$^+$ cells, we checked for Nimrod protein (P1 antibody, see details in 'Materials and methods') in Lz$^+$ sessile cells. It is worth noticing that, because crystal cells burst after bleeding (*Bidla et al., 2007*), there is a bias in immunofluorescence stainings in favor of more immature crystal cells. Contrarily to the lymph gland, the majority of Lz$^+$ cells are also Nimrod$^+$ (*Figure 2A–A″*). This result suggests that Lz$^+$ cells differentiate from a pool of mature plasmatocytes.

Interestingly, it has been shown before that plasmatocytes transdifferentiate into lamellocytes (*Honti et al., 2010*; *Meister and Ferrandon, 2011*). In this case, cells that are phagocytically active become non-phagocytic and start to express lamellocyte markers (*Honti et al., 2010*). Given this, we proceeded to test if the observed transition between Lz$^-$ and Lz$^+$ corresponds to a change of fate from plasmatocyte (phagocytic) to crystal cell (non-phagocytic). From our results, it is evident that phagocytosis index is higher in Lz$^-$ hemocytes but it is still non-negligible in Lz$^+$ hemocytes (*Figure 2B*). When we characterize cell area and GFP intensity in Lz$^+$ cells, it is possible to observe that the smaller and GFP$^{low}$ expressing cells are able to phagocyte bacteria (*Figure 2B'*). Large cells that have GFP$^{high}$ expression are virtually non-phagocytic cells. This indicates that induced Lz$^+$ cells are plasmatocytes with phagocytic activity, which is lost as they mature into crystal cells. Altogether, these results support the transdifferentiation of mature phagocytically active plasmatocytes into non-phagocytic crystal cells.

## Serrate expression in plasmatocytes is essential for crystal cell differentiation

As mentioned above, crystal cell numbers are reduced in larvae raised at a restrictive temperature in a thermo sensitive *Notch* allele background (*Duvic et al., 2002*). This reduction is visible in sessile hemocytes and in the lymph gland (*Duvic et al., 2002*; *Lebestky et al., 2003*). In the lymph gland *Notch* signaling has a cell autonomous role on Hml$^+$Lz$^-$ cells (*Mukherjee et al., 2011*). To test for the role of the Notch pathway in the differentiation of crystal cells in the clusters, we used RNAi exclusively in hemocytes by way of the HmlΔGAL4 driver. Firstly, we establish that Notch downregulation reduces the number of sessile crystal cells both in females (*Figure 3A*) and males (*Figure 3—figure supplement 1*). As mentioned above, in this experiment it is possible to observe that sessile crystal cell numbers are higher in females (compare controls in *Figure 2A* and *Figure 3—figure supplement 1*). In *Drosophila*, Notch is activated by two different ligands: Serrate and Delta (*Fiúza and Arias, 2007*). Only *Serrate* mutants have reduced numbers of embryonically derived crystal cell (*Duvic et al., 2002*; *Lebestky et al., 2003*). Similarly, knocking down *Serrate* in hemocytes (HmlΔ-GAL4>UAS-*Serrate* RNAi) reduces the number of crystal cells to a similar level than found using *Notch* RNAi, as opposed to disrupting Delta function (*Figure 3A*). This indicates that Serrate, the ligand necessary to induce Notch signaling in hemocytes, interestingly, it is expressed in the hemocytes themselves.

Importantly, the knockdown of Notch does not disrupt the hemocyte clusters nor changes the concentration of hemocytes in circulation (*Figure 3—figure supplement 2,3*). In the lymph gland, *Notch* activation is essential to induce *lozenge* upregulation (*Lebestky et al., 2003*). As observed in the lymph gland (*Terriente-Felix et al., 2013*), we can detect Notch enhancer GFP reporters in Lz$^+$ sessile hemocytes and, using video analysis, we can observe induction of *lozenge* in Notch activated cells (*Figure 3B*). To confirm that *Notch* knockdown inhibits the induction of *lozenge* in sessile

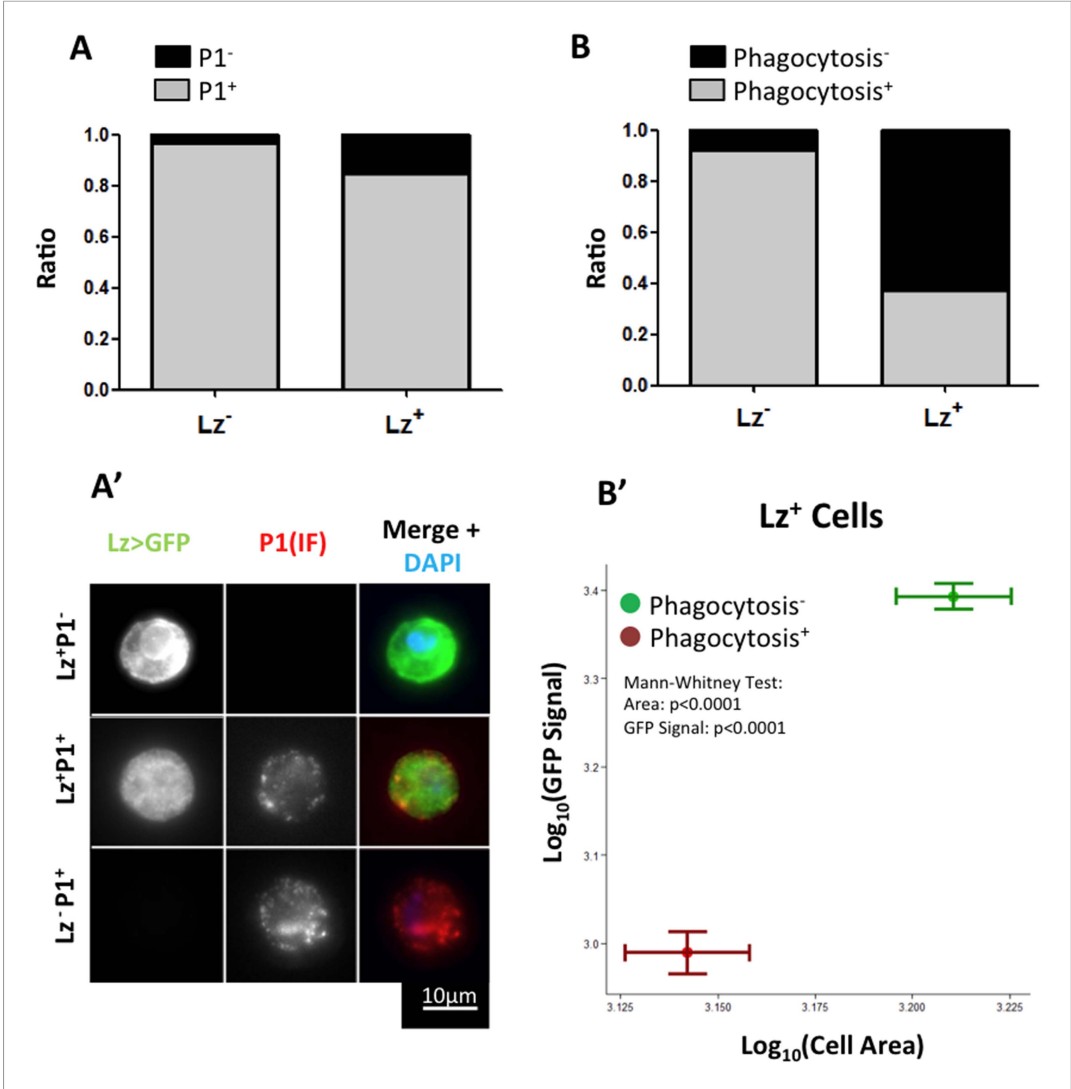

**Figure 2**. Lz$^+$ cells derive from mature plasmatocytes. (**A**) P1 immunofluorescence (IF) staining of sessile hemocytes marks the majority of Lz$^-$ and Lz$^+$ cells. Bars represent the mean ratio of P1$^+$ and P1$^-$ cells in these two population of cells (n = 10 samples) (**A'**) Examples of P1$^+$ Lz$^-$ plasmatocyte, P1$^+$ Lz$^+$ crystal cell and P1$^-$Lz$^+$ crystal cell. (**B**) Part of the Lz$^+$ cells are capable of doing phagocytosis. Bars represent the mean ratio between phagocytic and non-phagocytic cells (n = 5 samples). (**B'**) Phagocytic capacity in Lz$^+$ cells correlates negatively with both cell size and GFP intensity (measured by mean grey value of the picture). Points represent the mean and error bars represent the SEM.

hemocytes and not the maturation of Lz$^+$ cells into crystal cells, we measured the proportion of Lz$^+$ cells with anti-Lozenge antibody while inhibiting *Notch* expression in all hemocytes (HmlΔ-GAL4>*Notch*RNAi). The proportion of Lz$^+$ cells in this case is clearly reduced (*Figure 3C*). Because increased crystal cell apoptosis could also explain the reduced crystal cell numbers, we estimated hemocyte apoptosis upon Notch knockdown and could not find any significant difference to controls (*Figure 3—figure supplement 4*). Altogether these results confirm that Notch activation is essential to induce *lozenge* expression in larval hemocytes that will mature into crystal cells.

Since we used the HmlΔ-GAL4 driver, we could not distinguish if *Serrate* is necessary in Hml$^+$Lz$^-$, in Hml$^+$Lz$^+$, or in both cell types. To test these alternatives, we performed knockdown of Serrate with the Lz-GAL4 driver and found no reduction in the number of crystal cells (Females in *Figure 3D*, males in *Figure 3—figure supplement 5*). In addition, two other GAL4 drivers expressed in plasmatocytes, Eater-GAL4 and Pxn-GAL4, reduce the number of sessile crystal cells when driving Serrate RNAi

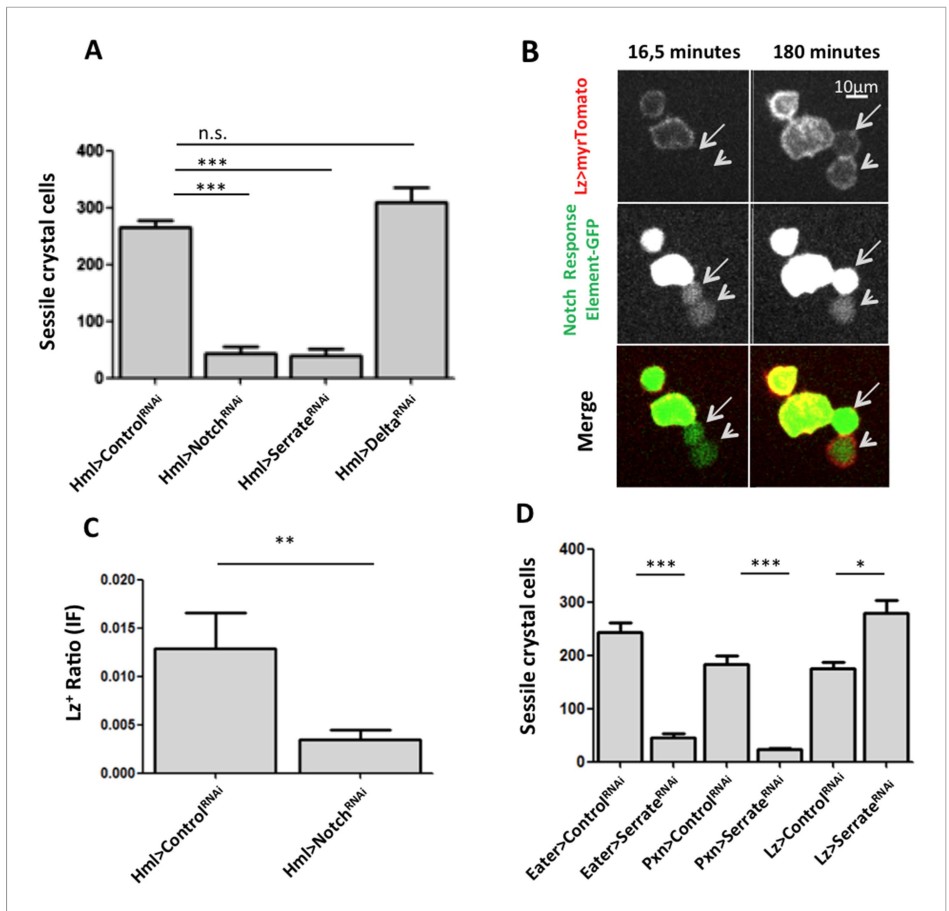

**Figure 3**. Serrate downregulation in plasmatocytes leads to a reduction in sessile crystal cell number. (**A**) Notch RNAi driven in all hemocytes reduces the number of melanized sessile crystal cells observed upon heat shock treatment to the whole larva. A similar level of reduction is seen with Serrate[RNAi] but not with Delta[RNAi] (n = 20). (**B**) Still images of a 3-hr video showing the induction of lozenge reporter expression in Notch activated hemocytes (arrows). (**C**) Notch[RNAi] reduces the proportion of Lz[+] cells in sessile hemocytes quantified with P1 immunofluorescence (IF) staining (n = 11 samples). (**D**) Serrate RNAi driven only in Lz[+] cells does not reduce the number of melanized sessile crystal cells seen upon heat shock treatment contrarily to two other drivers expressed in plasmatocytes, Eater-GAL4 and Pxn-GAL4 (n = 20 samples). In all graphics only female larvae are shown, error bars represent SEM, n.s. = non significant p-value, **p < 0.01, ***p < 0.001.

The following figure supplements are available for figure 3:

**Figure supplement 1**. Notch[RNAi] and Serrate[RNAi] but not Delta[RNAi] driven in all hemocytes reduce the number of melanized sessile crystal cells observed upon heat shock treatment to the whole larva (males are shown, n = 20).

**Figure supplement 2**. The localization of hemocyte in sessile clusters is not affected by Notch pathway manipulation through RNAi induction under HmlΔ-GAL4 control.

**Figure supplement 3**. Notch pathway manipulation through RNAi induction under HmlΔ-GAL4 control, does not change hemocyte concentration in hemolymph.

**Figure supplement 4**. Notch knockdown through RNAi under HmlΔ-GAL4 control does not increase cell death as measured by a flow cytometry Propidium Iodide (PI) exclusion assay , error bars represent the SEM. n.s. = p ≥ 0.05.

**Figure supplement 5**. Serrate RNAi driven only in Lz[+] cells does not reduce the number of melanized sessile crystal cells seen upon heat shock treatment contrarily to two other drivers expressed in plasmatocytes, Eater-GAL4 and Pxn-GAL4 (males are shown, n = 20 samples).

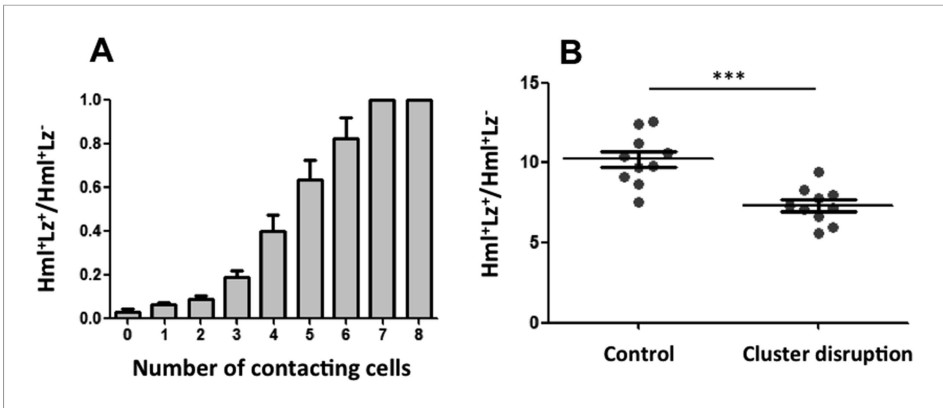

**Figure 4**. Cluster structure is necessary for crystal cell development. (**A**) Sessile hemocytes in Lz>mCD8GFP HmlΔ-cytoplasmic DsRed larvae were scored for the number of contacts. Probability of a cell being Lz$^+$ increases linearly with the number of cells it is in contact with. (n = 8) (**B**) In early third instar larvae, the continued disruption of clusters for a 10-hr period leads to a reduction in the proportion of Lz$^+$ cells (circulating and sessile cells were quantified). Error bars represent SEM, ***p < 0.001.

The following figure supplements are available for figure 4:

**Figure supplement 1**. In HmlΔ-nuclearDsRed; Lz-GAL4, UAS-GFP larvae, the dorsal cluster in the A7 segment is easily observed (left panel).

**Figure supplement 2**. Flow cytometry analysis for cell viability and Dot-GAL4 lineage tracing in cluster disrupted larvae.

**Figure supplement 3**. It is possible to detect lymph gland derived hemocytes by flow cytometry with Dot-GAL4 lineage-traced hemocytes in pupa (blue line) or when larvae are infected with parasitoid wasps (green line).

(females in *Figure 3D*, males in *Figure 3—figure supplement 5*). Hence, we conclude that plasmatocyte are responsible for *Serrate* signaling to activate *Notch* in other plasmatocytes and start crystal cell development.

## Hemocyte cluster structure is necessary for Notch-dependent larval hematopoiesis

For *Notch* pathway activation, cells need to be in contact because the ligand *Serrate* is membrane bound (*Guruharsha et al., 2012*). The fact that *Serrate* expression in Hml$^+$Lz$^-$ hemocytes is necessary for crystal cell development suggests that Lz$^+$ cells are induced when hemocytes are in contact within the clusters. If this is the case, we can predict that the probability of an Hml$^+$ cell to be also Lz$^+$ increases with the number of cell contacts it establishes with Hml$^+$Lz$^-$ cells. We quantified the number of cell contacts that each cell type makes within the sessile clusters and compared cells within the same size range (i.e., limited to the maximum size of Hml$^+$Lz$^-$ cells). As predicted, the proportion of Hml$^+$Lz$^+$/ Hml$^+$Lz$^-$ increases with the number of Hml$^+$Lz$^-$ cells with which it is in contact (*Figure 4A*).

Given that the clustering of hemocytes is important for hematopoietic decisions, we tested this idea further by assessing how the absence of clusters would affect crystal cell differentiation. A misexpression screen identified some genes that when overexpressed in hemocytes can disrupt hemocyte clusters (*Stofanko et al., 2008*). However, we did not use these lines to test for crystal cell/plasmatocyte ratio alterations because it would be difficult to discern between the effect of not having the clusters and the effect of gene up-regulation in hemocytes. We opted to disrupt the hemocyte clusters by manipulating physically the larvae (*Makhijani et al., 2011*). By rolling groups of larvae between two cover slides, it is possible to force the hemocytes to detach from the epidermis and enter hemolymph circulation. After cluster disruption, hemocytes start to aggregate again, gradually (*Makhijani et al., 2011*) (*Figure 4—figure supplement 1*). To maintain hemocytes in circulation for a period of 10 hr, we disrupted hemocyte clusters in larvae every 1 hr 30 min

(see 'Materials and methods'). Using flow cytometry, we measured the proportion of Hml$^+$Lz$^+$/Hml$^+$Lz$^-$ cells at the end of this treatment. The relative number of Hml$^+$Lz$^+$ cells decreases in the treatment group (*Figure 4B*), indicating that clusters are necessary for crystal cell differentiation.

However, the cluster disruption treatment could also disrupt the lymph gland and/or change the rate of apoptosis differentially between Hml$^+$Lz$^+$ and Hml$^+$Lz$^-$ cells. We tested both possibilities. Firstly, using a PI exclusion assay, we determined if the rate of apoptosis changes upon cluster disruption. There is no significant difference between control and treatment groups (*Figure 4—figure supplement 2*). Secondly, to check if cluster disruption affects the lymph gland, we made use of DotGAL4, a driver expressed in lymph gland hemocytes but not in circulating and sessile hemocytes (*Honti et al., 2010*). Through lineage tracing analysis with Dot-GAL4 driver, we can check cells that are derived from the lymph gland. Indeed, using flow cytometry, we can observe lymph gland-derived hemocytes in the circulating pool of wasp-infected larvae and pupae (*Figure 4—figure supplement 3*). When the same technique is used in cluster-disrupted larvae, there is no detectable lymph gland-derived hemocytes in circulation (*Figure 4—figure supplement 2*). Thus, the integrity of the cluster is necessary for crystal cell differentiation. Strikingly, in agreement with our observations, recent work shows that in *Eater* mutants that do not form hemocyte clusters, sessile crystal cells are absent (*Bretscher et al., 2015*).

Altogether these results support that *Notch* signaling is necessary for crystal cell differentiation and depends on cluster structure. It is important to note, though, that hemocytes form clusters of different sizes and at different locations along the larval body (Kurucz et al., 2007; *Makhijani et al., 2011*). Our cluster disruption procedure affects sessile hemocytes indiscriminately such that we cannot determine the relative importance of these cluster features on crystal cell differentiation.

## Discussion

Our motivation to carry out this work was to explain the increase of circulating and sessile crystal cell numbers during *Drosophila* larval development. This phenomenon is lined with an apparent paradox: no mature crystal cell has been seen dividing during larval stages (*Rizki, 1957*; *Lanot et al., 2001*; Krzemien et al., 2010) and crystal cells in the lymph gland do not enter circulation in homeostatic conditions (*Holz et al., 2003*). Crystal cell number increase may rely upon a population of pro-crystal cells that proliferates in the larva before cell maturation or that that exists in sufficient numbers at the beginning of development to mature into crystal cells throughout development. The first known upregulated gene diagnostic of crystal cell development is the transcription factor *lozenge* (Lz) (*Lebestky et al., 2000*). That is, a cell will be Lz$^+$ before it matures into crystal cell and maintains this expression upon differentiation (*Lebestky et al., 2000*). We have shown that throughout third instar development, the number of Lz$^+$ cells in the sessile population increases. This observation excludes the possibility that a population of Lz$^+$ cells exists in fixed number and matures into crystal cell. Lz$^+$ cells have been reported to proliferate during embryogenesis (*Lebestky et al., 2000*). Surprisingly, we do not see Lz$^+$ cell division in our video analysis. Although, with our results, we cannot exclude that a small proportion of Lz$^+$ cells proliferate during larval stages, we can show that *lozenge* induction in Hml$^+$Lz$^-$ is sufficient to explain the increase in Hml$^+$ Lz$^+$ cells during third instar larva.

The activation of *lozenge* in hemocytes is Notch-dependent with Serrate acting as the ligand (*Duvic et al., 2002*; *Lebestky et al., 2003*). When we ablate *serrate* expression only in hemocytes using the HmlΔ-GAL4 driver coupled to a UAS-Ser$^{RNAi}$, the number of differentiated crystal cells is severely reduced within the sessile population. This indicates that hemocytes are the cells responsible for crystal cell induction in sessile clusters. Moreover, the hemocytes inducing crystal cell development are themselves Lz$^-$ because *Serrate*$^{RNAi}$ driven by Lz-GAL4 driver does not decrease the number of crystal cells. For Notch to be activated it requires that a *Serrate* expressing cell is in contact for a certain period of time (*Guruharsha et al., 2012*). We show that this contact is a property of the clusters where Hml$^+$Lz$^+$ cells are induced from Hml$^+$Lz$^-$ cells. This observation establishes an important parallel between sessile and lymph gland crystal cell development. In both cases, the precursor to a crystal cell is an Hml$^+$Lz$^-$ hemocyte (*Mukherjee et al., 2011*). However, there is a fundamental difference between these two hematopoietic events regarding the cells where *lozenge* is first expressed. Although hemocytes only activate *lozenge* expression in the cortical zone of the lymph gland (*Lebestky et al., 2000*), the work of Krzemien et al. suggest that in the medullary zone of the lymph gland cells are already committed to become crystal cells at late second instar (Krzemien et al., 2010). This suggests that medullary zone cells migrating to the cortical zone can be considered pro-crystal

cells. In line with this observation, in the lymph gland, no co-localization between plasmatocyte marker P1 and crystal cell marker Lz is ever observed (*Terriente-Felix et al., 2013*; *Ferguson & Martinez-agosto 2014*). In contrast, our analyses in hemocyte clusters suggest that *lozenge* induction occurs in mature plasmatocytes. Firstly, they derive from P1 cells. Secondly, $Lz^+GFP^{low}$ cells can phagocyte as opposed to $Lz^+GFP^{high}$ cells, which loose this capacity. $Lz^+GFP^{low}$ cells are, according to our video analysis, the initial stages of crystal cell differentiation as virtually all $Lz^+GFP^{low}$ become $Lz^+GFP^{high}$. Throughout time cells increase their GFP expression and become larger. Altogether our results suggest that mature plasmatocytes can differentiate into crystal cells.

This conclusion may help explaining some disparate results in the literature. Firstly, circulating crystal cells in the larva derive from cells that express the plasmatocyte-specific marker *croquemort* during the embryonic stage (*Franc et al., 1999*; *Honti et al., 2010*). Secondly, Lebestky et al. consider that a small fraction of the Lz-GAL4 positive cells gives rise to plasmatocytes defined by morphology and *croquemort* expression (*Lebestky et al., 2000*). In light of our results, we propose that the plasmatocyte-like cells expressing *lozenge* are plasmatocytes in route to become crystal cells. To our knowledge, the hypothesis that plasmatocytes give rise to crystal cells was put forward by Rizky in 1957, with the purpose of explaining how crystal cells increase in number without proliferation (*Rizki, 1957*). Our work provides the first body of evidence that puts this idea to test and validates this hypothesis.

It is possible that, contrary to the lymph gland, hemocyte clusters are not regionalized structures (*Honti et al., 2014*). Yet, with the results shown here, we propose that hemocyte clusters work as a true hematopoietic tissue. Their presence and integrity are necessary for the proper establishment of $Hml^+Lz^+/Hml^+Lz^-$ numbers during larva development. Interestingly, the hemocytes in clusters are in dynamic relation with circulating hemocytes (*Babcock et al., 2008*; *Welman et al., 2010*; *Makhijani et al., 2011*). This is confirmed in our videos where cells can be observed entering circulation from the patches and leaving circulation to become sessile. This dynamics opens the possibility of a more complex number and cell type regulation mechanism operating at the whole-organism level. Secondly, another interesting property of sessile plasmatocytes consists of their higher division rate with respect to their circulating counterparts (*Makhijani et al., 2011*). This could happen because there is a different molecular 'environment' in hemocyte clusters (*Makhijani et al., 2011*) and/or because a sessile cell has an increased probability of entering cell division. We argue that the existence of these two characteristics, cell proliferation control and cell differentiation, is sufficient to consider the hemocyte clusters as hematopoietic tissues. In brief, hemocyte clusters enhance hemocyte proliferation and provide structure as to guarantee the necessary cell contacts that engage the signaling events behind cell fate decisions. Noticeably, hemocytes in clusters can be mobilized to circulation upon immune challenge (*Zettervall et al., 2004*), a process that is in part dependent on the small GTPase *Rac1* (*Xavier and Williams, 2011*). The role of hemocyte clusters is most likely restricted to larval stages because, once pupariation starts, a peak of ecdysone promotes the dispersion of hemocytes throughout the epidermis (*Regan et al., 2013*).

The differentiation of crystal cells from plasmatocytes within sessile clusters creates, in our view, an interesting parallel with macrophage development in vertebrates. Macrophages are the most plastic cells in the vertebrate's hematopoietic tissue and their specialization in vivo depends on the local microenvironment provided by the tissue they colonize (*Ostuni and Natoli, 2011*; *Wynm et al., 2013*). Similarly, here we show that in *Drosophila* larvae the microenvironment provided by hemocyte clusters is necessary to induce crystal cell differentiation from plasmatocytes, namely through a cell-contact mechanism involving Notch-Serrate.

A putative important difference between hemocyte clusters and the lymph gland concerns the mechanisms in control of cell proliferation and differentiation. In support of this notion, misexpression of some genes in hemocytes can disrupt hemocyte clusters without affecting lymph gland morphology (*Stofanko et al., 2008*). Tightly linked to this question is one other fundamental aspect that remains unaddressed: the control of proportions between different cell types. Throughout homeostatic development, it is commonly observed, both in vertebrates (*Almeida et al., 2005*) and invertebrates (*Rizki, 1957*), that blood cell types respect fixed relative numbers. Also, it is now evident that plasmatocytes are very plastic cells and may represent a rare case of functionally mature cells transdifferentiating into other cell types: lamellocytes (*Honti et al., 2010*) and crystal cells. Transdifferentiation, the process where a cell changes its cell fate without passing through a less differentiated state, is used recurrently in cell culture assays but rarely seen in vivo (*Jopling*

*et al., 2011*). How recurrent this mechanism may be in animal development presents itself as one important question for the future. We consider that acknowledging this novel hematopoietic organ, dynamically attached to the circulating hemocyte population and relying on structure-dependent signaling events to promote blood homeostasis, brings us a step closer to addressing these outstanding fundamental questions of *Drosophila* hematopoiesis.

# Materials and methods

## Fly stocks and parasitoid maintenance

All fly stocks were maintained in standard fly food at room temperature. Experiments were performed at 25°C except for RNAi experiments that were performed at 29°C. The following stocks were obtained from the Bloomington Stock Center: Lz-GAL4 UAS-mCD8GFP (#6314); Lz-GAL4 UAS-GFP (#6313); Notch Responsive Element (#30727); UAS-FLP UbiFRTSTOPStinger (#28282); UAS-myrtdTomato (32221); HmlΔGAL4 UAS-EGFP (30140). The following stocks were obtained from the Vienna *Drosophila* Resource Center: CG9313$^{RNAi}$ (#103600), Notch$^{RNAi}$ (#100002), Serrate$^{RNAi}$ (#108348), Delta$^{RNAi}$ (#109491) (*Dietzl et al., 2007*). The line Cg9313$^{RNAi}$ was used as control for RNAi experiments since it is a gene specifically expressed in male testis (Paulo Navarro, personal communication). The line HmlΔ-nuclearDsRed was a kind gift from Marc Dionne (*Clark et al., 2011*). The line HmlΔ-DsRed was a kind gift from Utpal Banerjee (*Makhijani et al., 2011*). The Eater-GAL4 (II) was a generous gift from Robert A Schultz (*Tokusumi et al., 2009*). *Leptopilina boulardi* females of the strain G486 (a kind gift from Fernando Roch) were allowed to lay eggs on second instar *Drosophila Dif* mutants at room temperature. Adult parasitoids were maintained in fly food vials with a drop of honey.

## Larva staging

Around 20 female flies were placed in a cage with a food plate containing yeast. Egg lays took place at 25°C for 6 hr. At ~72 hr midpoint after egg lay, second instar larvae were selected based on spiracle morphology and transferred into a new food plate. After 2 hr, larvae that molted into third instar were selected and transferred into a food tube. This first time point is referred to as 2 hr after third instar.

## Flow cytometry analysis and cell viability assay

Larvae were bled in 200–400 μl of Ringer's solution. The number of larvae greatly depends on the experiment but at least 10 larvae were used in each sample. The hemocyte dilution was passed through a 30-μm filter to exclude cell aggregates. The samples were maintained on ice until acquisition. Acquisition was performed in CyAn ADP cytometry Analyzer (DAKO Cytomation, Beckman Coulter) with Summit software (DAKO). Hemocyte population was gated in Forward Scatter (FSC) and Side Scatter (SSC) channels and single events in FSC and Pulse Width channels. GFP and DsRed were measured in the appropriate channels. To analyze results, it was used the Flowing Software (version 2.5.0). To analyze cell viability a stock solution of Propidium Iodide (PI) was diluted in 200 μl of Ringer's solution to a final concentration of 2 μg/ml. Positive events for PI were considered dead or dying cells.

## Video preparation and analysis

Male larvae were selected as described above, briefly washed in Ringer's solution, dried on filter paper and attached to double-sided tape on a cover slip. A second cover slip is placed on top of the larva (dorsal side) so that the larva is stuck between two cover slips. The larva does not move but stays alive for at least 12 hr in a humid chamber. The pressure from the cover slip affects the A7 dorsal cluster, most probably because of the disruption of normal hemolymph circulation. Hence, we imaged more anterior epidermal clusters that were not so affected. The larva was mounted in an inverted confocal spinning microscope (Andor Revolution xD). The temperature of the slide chamber was maintained at 25°C and 95% relative humidity. A Z-stack of pictures ranging 28 μm was taken every 1 min 30 s for the GFP and RFP channel throughout a period of 3 hr. At the end of the video each larva was checked for viability by observing the beating dorsal vessel and mouthparts movement. Only one larva died during the process. Z-stacks were then analyzed manually in FIJI software (*Schindelin et al., 2012*). With video analysis, we estimated the Differentiation Proportion during 3 hr. To extrapolate the number of Hml$^+$Lz$^+$ induced cells for other time points, we used the formula:

$$Hml^+Lz^+_{t+1} = Hml^+Lz^+_{t0} + Hml^+Lz^-_{t0} \times (\text{Differentiation Proportion}).$$

## Total sessile hemocyte counts, total hemocyte load, and crystal cell counts

To count the total number of sessile hemocytes in all larvae, pictures were taken in four different angles, using the appropriate fluorescence markers for hemocytes, under a fluorescence stereoscope (Zeiss SteREO Lumar.V12). The last body segment of the larva was excluded because it is difficult to image. To estimate hemocyte concentration in the hemolymph, six wandering male or female larvae were briefly washed in Ringer's solution, dried in filter paper, pooled in a glass well and bled by rupturing the cuticle in the ventral side (to avoid disturbance of sessile hemocytes in the dorsal part where they are more abundant). The hemolymph was collected and pooled into a 0.5 ml microcentrifuge tube and diluted 1:10 with Ringer's solution. 9.5 µl of diluted hemolymph was loaded into a Neubauer chamber and hemocytes counted in squares of 1 mm$^2$. This way the hemocyte concentration can be estimated by the formula: [number of counted cells] $\times$ 10$^5$ cells/ml. To count sessile crystal cells in all larvae, we performed a 70°C heat shock treatment for 10 min (*Neyen et al., 2014*). With this treatment mature crystal cells melanize and it is possible to count sessile mature crystal cell numbers trough the cuticle.

## Hemocyte immunohistochemistry

To collect circulating hemocytes larvae were bled in Ringer's solution from the ventral side to minimize disruption of sessile clusters. To collect sessile hemocytes the dorsal part of the cuticle was dissected, washed twice in Ringer's solution and hemocytes were removed by gently passing the forceps in the cuticle. Hemocytes were allowed to settle to slide glass reaction well (Ø 5 mm Marienfeld, Lauda-Königshofen, Germany) in a humid chamber for 10 min and fixed with 4% formaldehyde solution for 20 min. After fixation cells were washed three times with PBS and blocked with PBST (PBS + 0.1% Triton X + 1% normal goat serum) for 30 min. After washing the cells with PBS, the primary antibody was added at the correct dilution and cells incubated overnight at 4°C. Cells were then washed three times with PBS for 15 min and the secondary antibody added in the correct concentration. Cells were incubated for 3 hr at room temperature or at 4°C overnight with the secondary antibody. The secondary antibody was washed three times with PBS. DAPI was added and incubated for 3 min followed by three washes with PBS. Slides were mounted with 80% glycerol solution and kept at 4°C before image acquisition. Secondary antibodies used: AlexaFluor 488 and AlexaFluor 546 (1:1000 dilution, Life Technologies, NY, USA). Primary antibodies used: anti-*lozenge* (1:100 dilution, Developmental Studies Hybridoma Bank, University of Iowa, US) anti-*NimrodC1* (P1 antibody, kind gift from Istvan Andó, 1:100 dilution). Some *D. melanogaster* lines have a deletion in *NimrodC1* locus and the epitope for P1 antibody is not present (*Honti et al., 2013*). Because we could not detect P1 staining on Lz-GAL4 UAS-mCD8GFP line hemocytes, we crossed it with Oregon R males and tested the F1.

## Cell imaging

Larvae were rolled to take pictures from different angles. Hemolymph smears with live or fixed cells were imaged in a Leica DMRA2 microscope coupled with a CoolSNAP HQ CCD camera.

## Phagocytosis assay

To test phagocytosis, early third stage larvae were injected with 69 nl of pHrodo Red *E. coli* BioParticles (1 mg/ml; Molecular Probes). Injected larvae were maintained in yeast for 1 hr before ~10 larvae were bled in 20 µl PBS (pH = 7.4). Hemocytes were allowed to settle for 20 min at room temperature in a humid chamber, washed with PBS and pictures were taken immediately.

## Cluster disruption assay

Pools of ~20 early third instar male larvae were transferred to fresh yeast on a plastic petri dish placed in a humid chamber. Every 1 hr 30 min, larvae were taken from yeast, cleaned in Ringer's solution and dried in filter paper. Groups of ~5 larvae were rolled several times by pressing a cover glass to disrupt hemocyte clusters. Control larvae were maintained on yeast. Before larva sampling the two groups were subjected to cluster disruption to sample both circulating and sessile hemocytes.

## Statistical analysis

Every experiment was repeated at least twice to check for reproducibility. Samples were tested for normality with Shapiro–Wilk test and the appropriate statistical test was then applied. Student's *t*-tests were used to compare two treatments. ANOVA (or Kruskal–Wallis as non-parametric test) was performed when several comparisons were necessary and Dunnett's multiple comparison tests (or Dunn's multiple comparison test) were applied to test differences between pairs of treatments. Statistical test and graphics were performed in Prism v5.01 (GraphPad Software) and R v2.15.2.

## Acknowledgements

We thank all members of the laboratories of Élio Sucena, Patrícia Beldade, and Christen Mirth for important discussions throughout this project. We are indebted to Bruno Lemaitre for sharing unpublished results. We thank Marc Dionne, Robert A Schultz, and István Andó for fly lines and/or reagents, the VDRC (Vienna, Austria) for the RNAi fly lines and the Hybridoma Bank (Iowa, US) for monoclonal antibodies. This work was supported by Fundação Calouste Gulbenkian/Instituto Gulbenkian de Ciência and by Fundação para a Ciência e a Tecnologia (SFRH/BD/51175/2010 to ABL).

## Additional information

### Funding

| Funder | Grant reference number | Author |
|---|---|---|
| Fundação para a Ciência e a Tecnologia | SFRH/BD/51175/2010 | Alexandre B Leitão |
| Calouste Gulbenkian Foundation | | Alexandre B Leitão, Élio Sucena |

The funders had no role in study design, data collection and interpretation, or the decision to submit the work for publication.

### Author contributions

ABL, Conception and design, Acquisition of data, Analysis and interpretation of data, Drafting or revising the article, Contributed unpublished essential data or reagents; ÉS, Conception and design, Analysis and interpretation of data, Drafting or revising the article, Contributed unpublished essential data or reagents

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
