## [Decision Letter]

Thank you for sending your work entitled “Drosophila sessile hemocyte clusters are true hematopoietic tissues that regulate larval blood cell differentiation” for consideration at eLife. Your article has been favorably evaluated by Janet Rossant (Senior editor), Utpal Banerjee (Reviewing editor), and 3 reviewers.

Several suggestions from the reviewers are fairly straightforward, but it is important to attend to them prior to acceptance. The Reviewing editor and the other reviewers discussed their comments before we reached this decision, and the Reviewing editor has assembled the following comments to help you prepare a revised submission.

In general, the manuscript represents careful analysis and is an important step forward in further defining the Drosophila hematopoietic system. The work also extends previous results from other laboratories in important and valuable ways. Please attend to the following issues:

1) The verdict on the post-mitotic character of crystal cells is still out. Perhaps there are stages of development when they can and other stages when they cannot divide. It will be good to point out the published data and confine your comments to sessile hemocytes. Also, as there are different classes of sessile hemocytes, all of which might get disrupted by physical force, perhaps soften the emphasis on which specific pool is affected. It is good enough to describe where a phenomenon is observed and point out that other areas will have to be investigated in the future.

2) No data are presented for earlier instars of development. The authors should provide these data or explain in the manuscript why they decided to look only at the third instar.

---

## [Author Response]

*In general, the manuscript represents careful analysis and is an important step forward in further defining the Drosophila hematopoietic system. The work also extends previous results from other laboratories in important and valuable ways*. *Please attend to the following issues:*

*1) The verdict on the post-mitotic character of crystal cells is still out. Perhaps there are stages of development when they can and other stages when they cannot divide. It will be good to point out the published data and confine your comments to sessile hemocytes. Also, as there are different classes of sessile hemocytes, all of which might get disrupted by physical force, perhaps soften the emphasis on which specific pool is affected. It is good enough to describe where a phenomenon is observed and point out that other areas will have to be investigated in the future*.

We agree with these critiques regarding the post-mitotic state of the crystal cells and the restriction of our conclusions to the sessile sub-population. We have addressed them in the text by changing a few sentences and including some precisions along the manuscript. Small adjustments were introduced in the Abstract and the following sentences were altered or included:

Introduction section, paragraph two: “On the other hand, all reports thus far concur in that mature crystal cells do not to divide during larval stages (Krzemien, Oyallon, et al. 2010; [28]; [40]) although they have been shown to proliferate during embryogenesis (29). Further characterization of a yet unknown source and undetermined mechanism of crystal cell differentiation is required to understand how do its number increases during larval development.”

Results section, paragraph two: “It has been suggested repeatedly that during larval stages mature crystal cells are post-mitotic (28; 40) making it reasonable to assume that new crystal cells differentiate as development proceeds.”

Results section, paragraph four: “This suggests that Hml^+^Lz^+^ sessile cells in dorsal clusters divide at an extremely low frequency or are post mitotic cells.”

Discussion section, paragraph one: “This phenomenon is lined with an apparent paradox: no mature crystal cell has been seen dividing during larval stages ([40]; [28]; Krzemien, Crozatier, et al. 2010) and crystal cells in the lymph gland do not enter circulation in homeostatic conditions (17).”

Results section, final paragraph: “It is important to note, though, that hemocytes form clusters of different sizes and at different locations along the larval body (Kurucz, Váczi, et al. 2007; [31]). Our cluster disruption procedure affects sessile hemocytes indiscriminately such that we cannot determine the relative importance of these cluster features on crystal cell differentiation.”

*2) No data are presented for earlier instars of development. The authors should provide these data or explain in the manuscript why they decided to look only at the third instar*.

We have no data for earlier larval stages and our analysis is indeed restricted to 3rd instar. This focus has both biological (the majority of hemocyte proliferation occurs in 3rd instar) and technical (the in vivo imaging at the heart of our work would not be feasible at earlier larval stages) justifications. To clarify this option and the concomitant limits of our observations and conclusions, we have modified the manuscript to clarify this point in the abstract and by adding:

Results section: “Here we focus on third instar larvae because this is the developmental window in which the majority of larval hemocytes are originated (28; 31). Moreover, at this stage we could develop reliable in vivo imaging procedures that render our analysis and interpretations more pertinent (see below).”